# Toll-like Receptor (TLR) Response in Chikungunya Virus Infection: Mechanism of Activation, Immune Evasion, and Use of TLR Agonists in Vaccine Development

**DOI:** 10.3390/vaccines13080856

**Published:** 2025-08-13

**Authors:** Mohammad Enamul Hoque Kayesh, Michinori Kohara, Kyoko Tsukiyama-Kohara

**Affiliations:** 1Department of Microbiology and Public Health, Faculty of Animal Science and Veterinary Medicine, Patuakhali Science and Technology University, Barishal 8210, Bangladesh; 2Transboundary Animal Diseases Center, Joint Faculty of Veterinary Medicine, Kagoshima University, Kagoshima 890-0065, Japan; 3Department of Microbiology and Cell Biology, Tokyo Metropolitan Institute of Medical Science, Tokyo 156-8506, Japan; kohara-mc@igakuken.or.jp

**Keywords:** chikungunya virus, Toll-like receptors, innate immunity, immune evasion, vaccine adjuvants

## Abstract

CHIKV is a re-emerging mosquito-borne arthritogenic alphavirus associated with large outbreaks and severe joint pain, and it poses a growing global health threat. Toll-like receptors (TLRs), as key pattern recognition receptors, detect viral components and initiate antiviral immune responses. Increasing evidence highlights the role of TLR signaling in shaping CHIKV infection outcomes, though its precise contribution remains unclear. CHIKV has developed mechanisms to evade host innate immune surveillance, promoting viral replication. TLR agonists show promise as vaccine adjuvants by enhancing immune responses. In this review, we summarize current insights into TLR-mediated immunity during CHIKV infection, the virus’s innate immune evasion strategies, and the potential of TLR agonists in improving vaccine efficacy.

## 1. Introduction

Chikungunya virus (CHIKV), the causative agent of human chikungunya fever, is a re-emerging mosquito-borne virus belonging to the genus *Alphavirus* and the family *Togaviridae* [1,2]. First identified in 1952 in Tanzania [3], CHIKV was associated with infrequent, localized outbreaks until 2004 [4]. Over the past two decades, however, the virus has spread extensively, with outbreaks reported in 114 countries across tropical and temperate regions [1,5]. The primary vectors responsible for transmission are *Aedes aegypti* and *Aedes albopictus* mosquitoes [2,6,7].

CHIKV poses a major global health burden, with an estimated 35 million infections annually, predominantly in Southeast Asia, Africa, and the Americas [8,9]. It has been designated a priority pathogen by the Coalition for Epidemic Preparedness Innovations and classified as an emerging infectious disease requiring urgent action by the World Health Organization [10]. Infection typically results in febrile illness accompanied by rash, polyarthralgia, and myalgia, and approximately 30% of patients may develop chronic arthralgia [1,11,12]. Severe complications such as neurological manifestations and fulminant hepatitis have also been reported [13]. Although mortality is rare, it primarily affects the elderly, infants, and immunocompromised individuals [14,15,16].

CHIKV is an enveloped, positive-sense, single-stranded RNA virus with an ~11.8  kb genome encoding two open reading frames (ORFs) [17]. The first ORF encodes four non-structural proteins (nsP1–nsP4), and the second ORF encodes five structural proteins: C, E3, E2, 6K, and E1 [18,19]. Three lineages are recognized based on geographic origin: East Central South African (ECSA) lineage, West African lineage, and Asian lineage [20,21].

Ixchiq (VLA1533), a live attenuated replicating vaccine, became the first chikungunya vaccine licensed by the U.S. Food and Drug Administration in November 2023 [5]. Vimkunya, a virus-like particle (VLP) vaccine, was the second approved in the U.S., based on favorable immunogenicity and safety data [22,23]. No specific treatments are currently available for CHIKV infection [24]. Vaccine effectiveness is limited by gaps in our understanding of the virus’s global burden, hindering efforts to assess and optimize its impact [8]. Although innate and adaptive immune responses are vital for viral control, they may also drive immunopathology [25,26]. Clarifying host immune responses is therefore essential for improving antivirals and vaccine strategies.

Toll-like receptors (TLRs) are key pattern recognition receptors (PRRs) of the innate immune system that detect conserved microbial structures, including microbe-associated molecular patterns and pathogen-associated molecular patterns (PAMPs), providing host protection [27,28,29]. Among PRRs such as TLRs, retinoic acid-inducible gene I (RIG)-I-like receptors (RLRs), nucleotide-binding oligomerization domain (NOD)-like receptors, C-type lectin receptors, AIM2-like receptors, and DNA-sensing receptors, TLRs are the most extensively studied and play a central role in recognizing viral nucleic acids and proteins [28,30].

Encoded by a large gene family, TLRs vary by species: humans have 10 functional TLRs (TLR1–TLR10), while mice contain 12 (TLR1–TLR9 and TLR11–TLR13) [31]. TLRs have distinct subcellular localization; TLRs 1, 2, 4, 5, 6, and 10 are surface-expressed and detect viral proteins, whereas TLRs 3, 7, 8, and 9 are localized to intracellular compartments such as the endoplasmic reticulum, endosomes, lysosomes, or endolysosomes, and recognize viral nucleic acids [32,33,34,35,36,37,38]. Specifically, TLR3 detects double-stranded RNA, TLR7/8 recognize single-stranded RNA, and TLR9 senses unmethylated CpG DNA [35,36,37,38].

Structurally, TLRs share a conserved architecture comprising an N-terminal ectodomain with leucine-rich repeats, a single transmembrane domain, and a cytosolic Toll/interleukin (IL)-1 receptor (TIR) domain [27,39]. The TIR domain induces signaling via adaptor proteins such as myeloid differentiation factor 88 (MyD88), MyD88 adaptor-like (MAL or TIRAP), TIR-domain-containing adaptor protein inducing interferon (IFN)-β (TRIF or TICAM1), TRIF-related adaptor molecule (TRAM or TICAM2), and sterile α- and armadillo-motif-containing protein, mediating specific downstream pathways [40,41,42]. MyD88 is associated with all TLR signaling pathways, except for TLR3 [43], which signals exclusively via TRIF. TLR3 and TLR4 activate the TRIF pathway, leading to IRF3 activation, while TLR4 uniquely involves both MyD88 and TRIF-mediated signaling [39].

TLRs are crucial for early pathogen recognition through sensing PAMPs, triggering proinflammatory cytokines and chemokines that prime the immune response and bridge innate and adaptive immunity [27,30,31,44,45]. However, dysregulated TLR activation can lead to immune-mediated pathology, highlighting the need for balanced TLR signaling [46,47,48,49,50,51]. In CHIKV infection, understanding TLR responses is crucial for elucidating disease pathogenesis and informing the development of vaccine adjuvants and host-directed antiviral therapies. Thus, a comprehensive understanding of TLR-mediated immune mechanisms is essential for therapeutic and preventive strategies. This review summarizes current knowledge on TLR responses to CHIKV, viral evasion mechanisms, and the potential of TLR agonists as adjuvants in CHIKV vaccines.

## 2. TLR Response to Chikungunya Virus (CHIKV) Infection

CHIKV infection induces both innate and adaptive immune responses, which are essential for controlling viral replication in the host [52]. As an RNA virus, CHIKV is recognized by PRRs, including TLRs (e.g., TLR3, TLR7, TLR8) and RIG-I-like receptors (RLRs, e.g., melanoma differentiation-associated gene-5 [MDA-5] and RIG-I), inducing the production of cytokines and IFNs [53,54]. In primary monocytes and monocyte-derived macrophages (MDMs), CHIKV activates TLR pathways, resulting in a type I IFN-dependent antiviral response and the early induction of proinflammatory cytokine production as soon as 6 h post-infection, with similar kinetics but varying protein levels [55]. The peak transcriptional expression of antiviral factors such as type I IFN, OAS1, and PKR occurs around 48 h post-infection. Notably, CHIKV infection induces the expression of TLR2, TLR7, and TLR8 in monocytes, and TLR3 and TLR7 in MDMs, highlighting a differential response between these cell types [55]. Furthermore, in MDMs, CHIKV infection enhances TLR1/2 signaling, leading to the activation of IRAK2 and NF-κB pathways and the subsequent secretion of TNF-α and IL-6 [56].

Elevated expression of TLR3 and its signaling adaptor protein TRIF have been reported in CHIKV-infected MDMs [56]. Several studies have demonstrated the antiviral role of TLR3 in CHIKV infection. Priya et al. demonstrated that the activation of TLR3 signaling along with the upregulation of TRAF6, TICAM-1, MCP-1, CXCL-10, IL-6, IL-4, ISG15, MX2, IFN-β, and OAS3 molecules could protect against CHIKV infection in mouse brain [57]. Defective TLR3 signaling has been shown to enhance CHIKV infection in vitro, and TLR3^−/^^−^ mice exhibited increased CHIKV viral loads, highlighting the antiviral role of TLR3 in CHIKV infection [58]. Treatment with poly(I:C), a TLR3 agonist, inhibited the replication of CHIKV in BEAS-2B (human bronchial epithelial cells) [59]. In another study, it was shown that poly (I:C) protected mice from CHIKV infection [57]. Overall, findings from different studies suggest that TLR3 plays a protective role in CHIKV infection.

Lani et al. reported that CHIKV infection induces the upregulation of TLR4 and TLR7 expression in Huh7 cells [60]. A previous study demonstrated that the activation of TLR4 leads to the phosphorylation of NF-κB, a key regulator of inflammation and infection outcomes [61,62]. Recently, Mahish et al. reported that TLR4 facilitates CHIKV attachment and entry into host macrophages through direct interaction with the viral E2 protein [63]. In line with this, both pharmacological inhibition using TAK-242 (Resatorvid) and the genetic ablation of TLR4 significantly reduced viral load and proinflammatory responses during CHIKV infection, highlighting TLR4’s critical role in mediating early viral entry and modulating host immune response [63]. Importantly, the observed reduction in viral load following TLR4 inhibition is consistent with its role in facilitating viral entry. Blocking TLR4 with TAK-242 likely impairs the virus’s ability to attach to and enter host cells, thereby decreasing the number of initially infected cells and ultimately lowering the overall viral burden. Mahish et al. further demonstrated that TAK-242-mediated reduction in CHIKV infection is associated with the decreased phosphorylation of p38 and SAPK-JNK signaling pathways, which function downstream of TLR4 activation [63]. Notably, time-of-addition experiments revealed that TAK-242 is most effective when administered before or during CHIKV infection, indicating that TLR4 plays a pivotal role in the early stages of infection, specifically during viral attachment and/or entry. In contrast, post-entry treatment with TAK-242 had no significant effect on viral RNA replication or E2 protein expression [63], underscoring that TLR4’s influence is primarily limited to the initial phase of infection rather than the post-entry replication stages.

Kashyap et al. investigated the expression patterns of TLR-induced cytokines and chemokines in chikungunya patients with and without neurological complications [64]. Elevated levels of cytokines such as IL-1β, IL-17A, and IL-8, along with chemokines including MCP-1, RANTES, IP-10, and thymus and activation-regulated chemokine (TARC), were observed in the sera of patients without neurological complications. In contrast, patients with neurological complications showed significantly higher levels of TNF-α, IFN-α, IL-6, IL-8, MCP-1, RANTES, monokine induced by IFN-γ (MIG), and TARC in cerebrospinal fluid [64]. These findings suggest a broad upregulation of TLR-mediated cytokine responses during CHIKV infection, particularly in cases with neurological involvement.

In acute CHIKV infection, elevated levels of IFN-γ, IP-10, MCP-1, and MIG correlate positively with disease severity, reflecting a strong proinflammatory cytokine response [65]. Notably, serum from patients infected with fast-replicating CHIKV isolates showed increased secretion of IFN-α, IL-1RA, IL-17F, IL-9, MCP-1, and MIP-1α, linking viral replication kinetics with heightened innate immune activation. During the convalescent phase, persistent arthralgia was associated with elevated IL-6, IL-1β, IL-9, and IP-10, alongside reduced levels of the anti-inflammatory cytokines IL-4 and IL-10 [65]. These findings indicate a robust and dynamic cytokine response closely linked to viral replication kinetics and disease progression, where sustained immune activation may contribute to chronic symptoms such as arthralgia [65,66]. A recent study further demonstrated that CHIKV infection triggers distinct cytokine profiles across disease stages: chronic chikungunya arthritis patients display elevated IL-1β, IL-6, and GM-CSF, with IL-1β identified as a potential biomarker, whereas recovered individuals show increased IL-12 and IFN-γ levels, suggesting roles in viral clearance and disease resolution [67].

RIG-I-like receptors (RLRs) also act as an important sensor for CHIKV detection. In an in vitro study, 5pppRNA, an agonist for RLR, was shown to dose-dependently inhibit CHIKV replication [68]. Priya et al. reported that CHIKV infection in neuronal cells activates TLR3 and TLR7 signaling pathways, while there was no induction of cytosolic RLRs such as RIG-I [69]. MDA5, however, was transiently upregulated at an early time point (12 h post-infection), but its expression declined as the infection progressed [69]. This transient expression may be partially attributed to the inherently low and tightly regulated expression pattern of RLRs in neuronal cells [70]. In addition, the preferential activation of TLR3 and TLR7 pathways in neuronal cells during CHIKV infection may reflect a tissue-specific adaptation that balances antiviral defense [71] with the need to limit inflammation and cytotoxicity [72]. This TLR-specific activation led to the IRF-7-dependent induction of type I IFNs, highlighting the prominence of endosomal TLRs in antiviral responses within the central nervous system. Notably, differential regulation of TLR signaling has been implicated in the increased virulence observed in mutant strains of CHIKV. Pretreatment with Poly I:C, a TLR3 agonist along with IFN-β and TNF-α, significantly inhibited CHIKV replication, underscoring the protective potential of modulating TLR-mediated innate immunity [69].

Single-nucleotide polymorphisms (SNPs) in TLR genes can alter receptor functions, impairing or enhancing ligand binding or receptor activation, which may lead to altered susceptibility to infection and disease outcome [73,74]. The G allele of the TLR7 variant (rs3853839 G/C) and the G allele of TLR8 (rs3764879 G/C) were associated with protection against CHIKV infection, suggesting the association of TLR7 and TLR8 genes in CHIKV infection [75]. Individuals with certain TLR7 and TLR8 gene SNPs (e.g., rs3853839, rs3764879, rs179010, rs5741880) showed a higher risk of CHIKV infection, suggesting that while TLR7/8 pathways provide an antiviral role, specific gene variants may compromise their protective role and facilitate infection [76]. These findings indicate that TLR7 and TLR8 polymorphisms may have pleiotropic effects in CHIKV infection, contributing to both host defense and pathogenesis based on the genetic background and immunological context, warranting further investigation into their underlying mechanisms.

A recent study demonstrated that interferon-induced transmembrane proteins (IFITM1, IFITM2, and IFITM3) inhibit CHIKV infection in human skin fibroblasts, and their overexpression significantly upregulates antiviral genes, including TLR3, TLR7, TLR8, and TLR9, along with key downstream signaling molecules such as TRADD, IRAK1, TRAF6, and MAP3K7 involved in TLR signaling pathways [77]. These findings underscore the critical antiviral role of IFITM1, IFITM2, and IFITM3 in inhibiting CHIKV infection as well as the activation of TLR genes in human skin fibroblasts. Different TLRs may respond differentially to infection, and an overall TLR response to CHIKV infection has been summarized in Figure 1. Acronyms used in Figure 1 are defined in the figure legend.

## 3. Inhibition of Innate Immune Response by Chikungunya Virus (CHIKV) Infection

Hosts possess defense mechanisms that act to prevent the establishment of infection; therefore, successful viral infection requires the inhibition or evasion of early host defenses mediated by the innate immune system. The Janus kinase-signal transducer and activator of transcription (JAK-STAT) signaling pathway is a key mediator of IFN signaling, and interactions between the JAK-STAT cascade and antiviral IFNs are critical to the host’s immune response against viral infection [78]. In a murine model, Geng et al. demonstrated that STING deficiency results in elevated viremia during the acute phase of CHIKV infection despite a normal-type I IFN response. In addition, STING-deficient mice exhibited exacerbated arthritis pathology, characterized by increased foot swelling, joint damage, and dysregulated chemokine expression, highlighting STING’s role in controlling viral replication and modulating inflammatory responses [79]. CHIKV capsid-mediated inhibition of cytosolic DNA sensing pathways like that of cyclic GMP-AMP synthase (cGAS) induction has been shown to reduce the antiviral efficacy of the cGAS-stimulator of interferon genes (STING) pathway [80]. CHIKV nsP1 can interact with STING, significantly inhibiting the IFN-β promotor activation induced by cGAS-STING [80]. CHIKV nsP1 catalyzes the capping of viral RNA to mimic host mRNA, thereby evading detection by innate immune sensors such as RIG-I and MDA5 [81,82]. Consequently, nsP1 may serve as a potential target for inhibiting CHIKV infection [83].

TANK-binding kinase 1 (TBK1) is a critical signaling intermediate downstream of various PRRs signaling, including TLRs and RLRs. It plays a pivotal role in antiviral innate immunity by activating interferon regulatory factor (IRF) 3, which in turn leads to the production of type I IFNs [84]. CHIKV-nsP2 was shown to block both type I and type II IFNs induced by JAK-STAT signaling [85]. A preprint study reported that CHIKV-nsP2 plays an important role in suppressing both TLR and RLR signaling pathways by blocking TBK1 and IRF3 proteins [86]. Moreover, CHIKV-nsP2 plays a critical role in inhibiting host transcription and mediating cytopathic effects, serving as a key determinant of CHIKV pathogenesis [87]. CHIKV nsP3 contributes to immune evasion by inhibiting NF-κB activation through its macro domain, which possesses ADP-ribosyl hydrolase activity; disruption of this function impairs viral infectivity, highlighting its role in modulating host antiviral signaling [88]. nsP3 interacts with host proteins’ nucleosome assembly protein 1 like proteins 1 and 4, supporting CHIKV replication [89]. CHIKV nsP4 blocks the phosphorylation of eIF2α, which prevents activation of the protein kinase RNA-like endoplasmic reticulum kinase (PERK) pathway in the host’s stress response, and sustains efficient replication, contributing to innate immune evasion [90].

CHIKV nsP2, E2 and E1 were reported to strongly suppress IFN-β promoter activation through the MDA5/RIG-I signaling pathway, suggesting their role as key antagonists of type I IFN induction [91]. Delineation of the underlying mechanisms is critical for virus-specific therapeutics and vaccine development [91]. The CHIKV E1 glycoprotein plays a critical role in modulating viral attachment and membrane fusion with host cells, thereby enhancing infectivity [92]. Mutations in E1, such as V156A and K211T, may also alter recognition by innate immune sensors, potentially contributing to early immune evasion, although this requires further investigation [92]. The interaction between E1 and E2 is critical for the proper assembly of the CHIKV glycoprotein complex [93]. CHIKV can also utilize the host protein to facilitate viral replication. For example, CHIKV nsP2 and nsP3 interact with host chaperone proteins HSP-90β or HSP-90α and promote viral replication [94].

RNA interference (RNAi) is a post-transcriptional gene regulation mechanism that forms part of the innate immune system, enhancing antiviral immune responses through multiple signaling pathways, including TLR signaling [95,96]. Cleavage products generated by RNAi, such as siRNAs or small single-stranded RNAs, can act as ligands for nucleic acid-sensing receptors, including TLR3 (dsRNA), TLR7/8 (ssRNA), and protein kinase R (PKR). The activation of these pathways can subsequently induce type I IFN and proinflammatory cytokine production [97]. CHIKV nsP2 and nsP3 proteins have been shown to exhibit an RNAi suppressor effect in the insect Sf21 RNAi sensor cell line, thereby facilitating viral replication and immune evasion [98]. Figure 2 illustrates the immune evasion strategies employed by CHIKV’s structural and nonstructural proteins, which interfere with innate immune signaling to suppress the host immune response and facilitate viral replication.

## 4. TLR Agonist as an Adjuvant for CHIKV Vaccine Development

Several vaccine platforms have been investigated for CHIKV, including live-attenuated, inactivated, mRNA, viral vector, and virus-like particle (VLP) vaccines, each with varying degrees of efficacy and safety [9,99,100,101]. Among these, live-attenuated and VLP-based candidates, such as VLA1553 and Vimkunya, have progressed to late-stage clinical trials, showing strong immunogenicity and favorable safety profiles [22,102]. Despite this progress, concerns related to safety and complex manufacturing processes continue to limit the widespread application of some platforms [99,100]. To address these challenges and enhance vaccine efficacy, modern adjuvants, particularly those based on TLR agonists, are being utilized to mimic PAMPs, thereby improving the magnitude and quality of adaptive immune response [103,104]. TLRs play a central role in dendritic cell maturation and antigen presentation, making TLR agonists especially valuable for vaccines that require enhanced immunostimulation, such as VLP and nucleic acid-based platforms. Notably, TLR-based adjuvants have demonstrated effectiveness in boosting immune responses in vaccines targeting various viruses, including hepatitis B and C viruses, HIV-1, influenza virus, and SARS-CoV-2 [46,105,106,107,108].

Currently, only two TLR agonists have received regulatory approval as vaccine adjuvants: monophosphoryl lipid A (MPLA), a TLR4 agonist, and CpG 1018, a synthetic TLR9 agonist. MPLA is a component of the AS04 adjuvant system, which combines MPLA with aluminum hydroxide that is used in the Cervarix vaccine, targeting human papillomavirus (HPV) types 16 and 18, and in HBV-AS04 vaccine, a recombinant hepatitis B formulation [109,110,111,112,113]. CpG 1018 is incorporated into HEPLISAV-B, a recombinant hepatitis B surface antigen (HBsAg) vaccine that has shown superior seroprotection, particularly in populations with reduced immune responsiveness such as older adults, individuals with diabetes, and those with chronic kidney disease, compared to Engerix-B [109,110]. While other TLR agonists (e.g., targeting TLR3, TLR5, TLR7/8) are under preclinical or clinical investigation, they are not yet approved for use in licensed human vaccines [111]. Nevertheless, the strategic incorporation of TLR agonists remains a promising approach for enhancing the immunogenicity and protective efficacy of CHIKV vaccines. However, this approach requires the careful selection of appropriate TLR agonists, rigorous validation of their safety and immunogenicity in preclinical models, and a comprehensive evaluation in clinical settings.

A recent study demonstrated that TLR4 agonist monophosphoryl lipid A (MPLA) enhanced the immunogenicity of an inactivated CHIKV vaccine, eliciting higher titers of neutralizing antibodies compared to the unadjuvanted formulation [114], thereby warranting further investigation. In silico analysis revealed that a multi-epitope vaccine candidate consisting of CHIKV structural proteins (E1, E2, 6 K, and E3) with TLR4 agonist was highly immunogenic and safe [115], requiring further investigation. Well-characterized TLR agonist adjuvants such as triacylated lipopeptides (e.g., Pam3CSK4) and their derivatives for TLR1/2, poly I:C for TLR3, bacterial lipopolysaccharide or MPLA for TLR4, bacterial flagellin for TLR5, imiquimod and resiquimod for TLR7/8, and CpG ODN for TLR9 [116] have demonstrated immunostimulatory potential in various vaccine platforms and warrant further investigation for their application in CHIKV vaccine development [117]. The use of multiple TLR agonists as adjuvants has been shown to enhance the immune response [118]; therefore, further studies are warranted to evaluate their potential in broadening the efficacy of CHIKV vaccine formulations.

## 5. Discussion

The significance of TLRs and RLRs in sensing CHIKV and initiating antiviral responses is becoming increasingly evident [77]. However, an inappropriate TLR response may support immunopathogenesis instead of providing protection to the host [69]. TLR3 plays an essential role in protecting CHIKV infection, as it was demonstrated that TLR3 agonist treatment inhibits CHIKV replication in vitro by upregulating IFN-β and other proinflammatory cytokines. However, another study also reported no difference in viral loads in tissues of CHIKV-infected wild type or TLR3^−/−^ mice [119]. These discrepancies may reflect differences in experimental models, the timing of immune responses, or compensatory mechanisms in vivo, and highlight the need for further investigation into the role of TLR3 in CHIKV pathogenesis. Although TLR7 and TLR8 are possible pathways to be activated for IFN-I production during CHIKV infection, to date, TLR7/8 has not been directly evaluated in reference to CHIKV replication; however, studies have found indirect evidence of its involvement [75,76]. Future research should investigate the interactions between CHIKV and key RNA-sensing TLRs—especially TLR3, TLR7, and TLR8—to better define their roles in host defense and viral pathogenesis.

During the early stages of viral infection, a complex network of innate immune factors is rapidly activated, with IFNs playing a central role in orchestrating antiviral defenses through the modulation of downstream signaling pathways. CHIKV has evolved multiple mechanisms to evade host immune responses, notably through the activity of its nsP2, which effectively suppresses IFN production and facilitates viral replication. Given its pivotal role in immune evasion, CHIKV-nsP2 represents a promising target for the development of antiviral therapeutics. Targeting the enzymatic functions of nsP1 could disrupt viral RNA stability and immune evasion mechanisms, offering a promising avenue for antiviral intervention [83]. Nevertheless, the precise functions of several structural and non-structural CHIKV proteins remain unclear. Elucidating their roles will be crucial for a comprehensive understanding of viral pathogenesis and may provide insights for the development of novel therapeutic interventions.

The appropriate selection of TLR agonist adjuvants is critical for formulating next-generation vaccines, which aim to induce an efficient adaptive immune response with minimal adverse reactions. However, such an approach requires careful selection of the TLR agonist, validation of its safety and efficacy in preclinical models, and a final evaluation in clinical settings. For CHIKV, TLR agonists hold promise as adjuvants by enhancing the immunogenicity of subunit or nucleic acid-based vaccine platforms. With further investigation, their capacity to modulate innate immune responses may prove particularly valuable in achieving durable and protective responses against CHIKV infection.

## 6. Conclusions

CHIKV continues to re-emerge globally, causing increasing morbidity. TLR responses play a pivotal role in CHIKV pathogenesis, and a deeper understanding of their involvement is crucial for unraveling host–virus interactions. Given that TLR signaling can exert both protective and pathological effects, achieving a balanced TLR-mediated response is essential for favorable disease outcomes. While TLR3 has been implicated in antiviral defense, the broader roles of TLRs in CHIKV immunity remain insufficiently defined. Similarly, although nsP2 is known to antagonize host defenses, the functions of other viral proteins in modulating innate immunity are still unclear and require further investigation. Advancing our knowledge of TLR–CHIKV interactions will support the development of targeted therapeutics and effective vaccines. TLR agonists hold promise as vaccine adjuvants, and future research should prioritize identifying the most suitable candidates for enhancing CHIKV vaccine efficacy.

## Figures and Tables

**Figure 1 vaccines-13-00856-f001:**
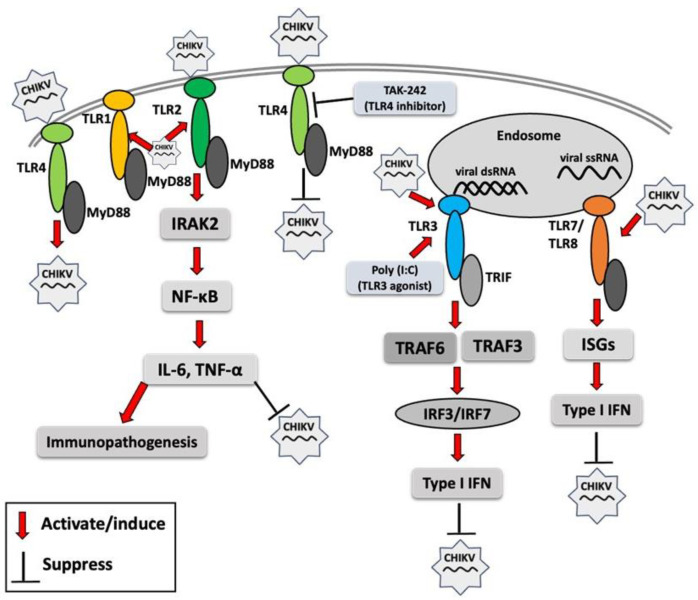
Toll-like receptor (TLR) response to chikungunya virus (CHIKV) infection. Red arrows indicate the activation of TLR signaling molecules by CHIKV or its components, resulting in enhanced or reduced CHIKV infection. MyD88 (myeloid differentiation primary response 88), TRIF (TIR domain-containing adaptor-inducing IFN-β), IRAK2 (interleukin-1 receptor-associated kinase 2), NF-κB (nuclear factor kappa-light-chain-enhancer of activated B cells), IL-6 (interleukin-6), TNF-α (tumor necrosis factor-alpha), poly I:C (polyinosinic/polycytidylic acid), TRAF3 (TNF receptor–associated factor 3), TRAF6 (TNF receptor-associated factor 6), IRF3 (interferon regulatory factor 3), IRF7 (interferon regulatory factor 7), ISG (interferon-stimulated gene), IFN (interferon).

**Figure 2 vaccines-13-00856-f002:**
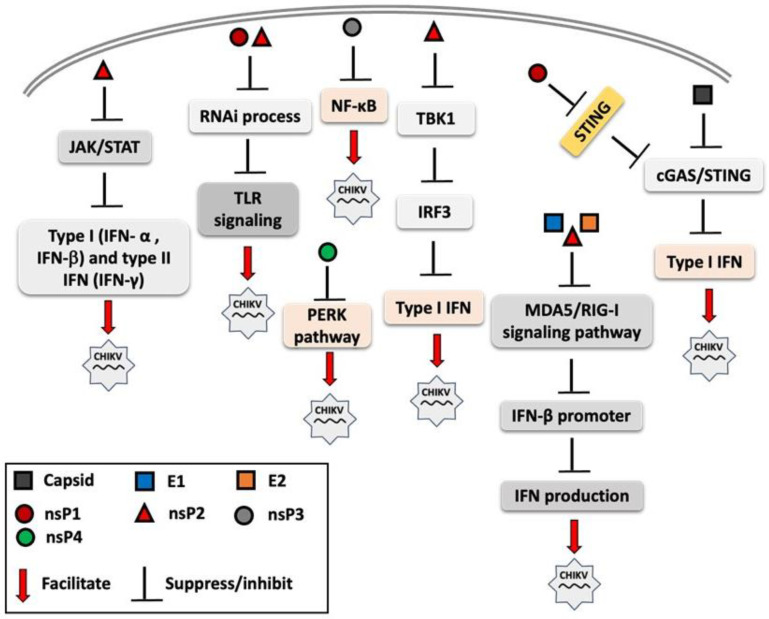
Overview of mechanisms by which chikungunya virus (CHIKV) proteins modulate and suppress host innate immune signaling. Both structural proteins (e.g., capsid and envelope proteins) and non-structural proteins (nsP1, nsP2, nsP3, and nsP4) interfere with pattern recognition receptor (PRR) signaling pathways, including Toll-like receptors (TLRs) and RIG-I-like receptors (RLRs) and other components of innate immune response. These viral proteins disrupt key steps in the interferon (IFN) response cascade, such as inhibiting interferon regulatory factor 3 (IRF3) activation, suppressing of Janus kinase-signal transducer and activator of transcription (JAK-STAT) signaling, and interfering with interferon-β promoter activation. The overall effect is the attenuation of antiviral responses, facilitating viral replication.

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
