# Peer review of "Toll-like Receptor (TLR) Response in Chikungunya Virus Infection: Mechanism of Activation, Immune Evasion, and Use of TLR Agonists in Vaccine Development"

_vaccines, 2025, doi:10.3390/vaccines13080856_

Round 1
Reviewer 1 Report
Comments and Suggestions for Authors
Mohammad Enamul Hoque Kayesh et al made a short review titled “Toll-like receptor (TLR) response in chikungunya virus infection: mechanism of activation, immune evasion, and use of TLR agonists in vaccine development”. Chikungunya spreads rapidly. The Southeast Aisa suffers from endemic cases annually. This review provides a timely update on the Chikungunya disease. The whole review is comprehensive but less focused.
Major concerns
- Two of the 13 pages only give a general introduction with less specific focuses, which provides information that is familiar to readers in the field. The introduction should be shortened substantially.
Author Response
Mohammad Enamul Hoque Kayesh et al made a short review titled “Toll-like receptor (TLR) response in chikungunya virus infection: mechanism of activation, immune evasion, and use of TLR agonists in vaccine development”. Chikungunya spreads rapidly. The Southeast Aisa suffers from endemic cases annually. This review provides a timely update on the Chikungunya disease. The whole review is comprehensive but less focused.
Major concerns
- Two of the 13 pages only give a general introduction with less specific focuses, which provides information that is familiar to readers in the field. The introduction should be shortened substantially.
Response: We appreciate the reviewer’s comments. In response, we have thoroughly revised the introduction to make it more concise and focused. Redundant or duplicate information has been removed to improve clarity and coherence.
Reviewer 2 Report
Comments and Suggestions for Authors
This review summarizes current insights into TLR-mediated immunity during
CHIKV infection, the virus’s innate immune evasion strategies, and the potential of TLR agonists in improving vaccine efficacy. There are some issues that should be addressed before acceptance:
1) In whole manuscript, the author presented the majority of cited studies were from the in vitro assays and murine models. There is limited discussion of how TLR expression influence the human immunity during CHIKV infection. Including this with related datasets or human studies would enhance translational relevance and help bridge the gap between bench and bedside.
2) In Section 3, the functions of other structural (capsid, E1, E2) and non-structural (nsP1, nsP3, nsP4) proteins are insufficiently described
Author Response
1) In whole manuscript, the author presented the majority of cited studies were from the in vitro assays and murine models. There is limited discussion of how TLR expression influence the human immunity during CHIKV infection. Including this with related datasets or human studies would enhance translational relevance and help bridge the gap between bench and bedside.
Response: Thank you for the valuable feedback. We agree with the reviewer’s suggestion and have now incorporated relevant findings from human studies that highlight the role of TLR expression and cytokine responses during CHIKV infection, thereby strengthening the translational relevance of the manuscript (lines 631-654).
2) In Section 3, the functions of other structural (capsid, E1, E2) and non-structural (nsP1, nsP3, nsP4) proteins are insufficiently described
Response: Thank you for your valuable feedback. We have revised Section 3 to include additional details on the functions of the structural proteins (capsid, E1, E2) and non-structural proteins (nsP1, nsP3, nsP4), highlighting their roles in immune evasion (lines 734-737, 746-753, 757-762).
The English could be improved to more clearly express the research.
Response: We have asked MDPI for English editing.
Reviewer 3 Report
Comments and Suggestions for Authors
The review manuscript entitled “Toll-like receptor (TLR) response in chikungunya virus infection: mechanism of activation, immune evasion, and use of TLR agonists in vaccine development”, reviews the data current on TLR-mediated immunity during CHIKV infection, innate immune evasion strategies, and the potential of TLR agonists in improving vaccine efficacy. CHIKV has been on the rise and more dangerous that it is mainly spread by mosquitoes creating health burden worldwide. The development of preventive measures, which include vaccines to combat the outbreak of CHIKV requires vigorous investigation and bringing together all available know materials together in order to find a solution. At this time when the population is becoming more vulnerable with diseases, this kind of reviews are important to summarize the data. The manuscript is clear and follows the idea from the beginning to the end. The introduction, the reported data and the conclusion are all supported by each other and the abstract. The references used in the review are relevant to the topic of discussions. However, there few comments to the authors that need to be addressed before the manuscript is published.
Broder comment
Lines 126-130: If TLR4 facilitates CHIKV attachment and entry into host macrophages, why does its inhibition reduce viral load. This information should be added and clearly stated to reduce contradiction.
RLRs is excluded in the neuronal cells activating of TLR3 and TLR7 signaling pathways. It will improve reading if the authors could add a sentence to explain why this might occur and why only the preference of RLRs.
Minor comments
Lines 143-149: how do the SNPs alters the function of TLR7/8. It will benefit reading if the authors could add one sentence explaining this.
The authors introduces “RNA interference (RNAi)”, what is the relevance of RNA interference (RNAi) to TLR signaling path way. It will benefit reading if the authors could add an explanation of this. For example, CHIKs suppression of RNAi further compromises immunity….
Lines 217- 223: at least the authors could add a sentence to show which vaccine platforms are more promising, just listing them lacks priority.
Which LTR agonists are clinically approved and used in different vaccines.
Author Response
Lines 126-130: If TLR4 facilitates CHIKV attachment and entry into host macrophages, why does its inhibition reduce viral load. This information should be added and clearly stated to reduce contradiction.
Response: Thank you for the comment. TLR4 facilitates CHIKV entry into macrophages, which contributes to viral replication and inflammation. Its inhibition reduces viral load by blocking entry and dampening the pro-inflammatory environment that supports viral propagation. We have clarified this in the revised text to address the contradiction (lines 461-463, 465-477, 628-630).
RLRs is excluded in the neuronal cells activating of TLR3 and TLR7 signaling pathways. It will improve reading if the authors could add a sentence to explain why this might occur and why only the preference of RLRs.
Response: Thank you for the suggestion. RLRs are minimally expressed in neuronal cells, which may limit their involvement in antiviral responses. Instead, TLR3 and TLR7 are more active in these cells due to their localization in endosomes, allowing efficient recognition of viral RNA. We have added a sentence to clarify this in the revised manuscript (lines 658-664).
Minor comments
Lines 143-149: how do the SNPs alters the function of TLR7/8. It will benefit reading if the authors could add one sentence explaining this.
Response: Thank you for the suggestion. We have added a sentence to clarify this in the revised manuscript (lines 672-674).
The authors introduces “RNA interference (RNAi)”, what is the relevance of RNA interference (RNAi) to TLR signaling path way. It will benefit reading if the authors could add an explanation of this. For example, CHIKs suppression of RNAi further compromises immunity….
Response: Thank you for the comment. We have added a sentence to clarify this connection in the revised manuscript (lines 767-777).
Lines 217- 223: at least the authors could add a sentence to show which vaccine platforms are more promising, just listing them lacks priority.
Response: Thank you for the suggestion. In line with reviewer comments, we have added a sentence to reflect prioritization of vaccine platform in the revised manuscript (lines 795-797).
Which LTR agonists are clinically approved and used in different vaccines.
Response: Thank you for the comment. Clinically approved TLR agonists used in vaccines include Monophosphoryl lipid A (MPLA), a TLR4 agonist used in Cervarix (HPV vaccine), and CpG 1018, a TLR9 agonist used in Heplisav-B (hepatitis B vaccine). We have added this information to the revised manuscript (lines 910-922).